



# Performance-based comparison of regionalization methods to improve the at-site estimates of daily precipitation

Abubakar Haruna[1], Juliette Blanchet[2], and Anne-Catherine Favre[1]

[1]Univ. Grenoble Alpes, Grenoble INP, CNRS, IRD, IGE, 38000 Grenoble, France
[2]Univ. Grenoble Alpes, CNRS, IRD, Grenoble INP, IGE, 38000 Grenoble, France

**Correspondence:** Abubakar Haruna (abubakar.haruna@univ-grenoble-alpes.fr)

**Abstract.** In this article, we compare the performances of three regionalization approaches in improving the at-site estimates of daily precipitation. The first method is built on the idea of conventional RFA (Regional Frequency Analysis) but is based on a fast algorithm that defines distinct homogeneous regions relying on their upper tail similarity. It uses only the precipitation data at hand without the need for any additional covariate. The second is based on the region-of-influence (ROI) approach in which neighborhoods, containing similar sites, are defined for each station. The third is a spatial method that adopts Generalized Additive Model (GAM) forms for the model parameters. In line with our goal of modeling the whole range of positive precipitation, the chosen marginal distribution model is the Extended Generalized Pareto Distribution (EGPD) on which we apply the three methods. We consider a dense network composed of 1176 daily stations located within Switzerland and in neighboring countries. We compute different criteria to assess the models' performances both in the bulk of the distribution as well as in the upper tail. The results show that all the regional methods offered improved robustness over the local EGPD model. While the GAM method is more robust and reliable in the upper tail, the ROI method is better in the bulk of the distribution.

## 1 Introduction

Flood events occurring at different time scales pose hazards that are of serious consequences to life and property. Reliable prediction of these events has always been a challenge, and is necessary for safe design and risk assessments. Simulation of these events is normally done via hydrological modeling which takes as inputs, among others, meteorological data such as temperature and precipitation. However, whatever the complexity of the model and how it represents the underlying hydrological behavior of the catchment, the accuracy, robustness and reliability of the flood predictions relies on the quality of the input data.

Precipitation intensities, the key input signal is normally modelled using probabilistic methods. Within this framework, a good probabilistic model should be able to predict rainfall intensities of any return level, both low, medium and extremes with reliable accuracy. Gamma distribution, a common choice over models such as log-normal, Weibull, exponential, fails in this aspect, as the tail is too light to model heavy intensities (Katz et al., 2002), and thereby resulting in underestimation of return levels (Naveau et al., 2016). Models based on the classical extreme value theory (EVT) are able to model the upper tail but one has to choose afterwards a model for the other intensities below a chosen threshold in the case of the Generalized Pareto



(GP) distribution for example. GP has been favored in hydrological applications (see e.g. Langousis et al., 2016) , and many
authors in the framework of modelling the full range, have considered stitching or a mixture of the GP tail and a light-tailed
density (e.g. Frigessi et al., 2002; Carreau and Bengio, 2009). Scarrot and MacDonald (2012) have reviewed these types of
models. This however, has the drawback of inflating the number of parameters to estimate (Naveau et al., 2016) and thus
complexify statistical inference. As an alternative, Naveau et al. (2016), proposed a model, an extension of the GP (afterwards
called EGPD), able to model adequately the entire range of positive precipitation, gamma-like in its lower tail, heavy tailed
(GP) in its upper tail, avoiding the need for threshold selection, and parsimonious by avoiding the use of mixtures. This model
has been used by many authors (e.g. Blanchet et al., 2015; Evin et al., 2018; Tencaliec et al., 2020; Gall et al., 2021).

   Although the EGPD uses all the data to estimate the parameters, the shape parameter which controls the upper tail behavior
is difficult to estimate based on few decades of data, the usual length of precipitation data, because there are few extremes
exhibiting much variability. Since precipitation is spatial by nature, several studies (Cunnane, 1988; Burn, 1990; Hosking and
Wallis, 2005) proposed the use of observations surrounding the local station in order to increase the quantity of data available
for estimation, thereby reducing the uncertainty involved in the estimation.

   Different methods exist in the literature to use information surrounding the station at hand (see Cunnane, 1988; Hosking
and Wallis, 2005). On one hand are the methods based on regional homogeneity (e.g. method of Hosking and Wallis (2005))
that involves pooling all observations, in hydrologically similar sites, in order to increase the sample size, and by so yielding
more accurate estimates of the parameters. Hydrologically similar sites are first defined using cluster analysis and then sub-
jected to some statistical homogeneity tests on the scaled observations. Thereafter, a chosen distribution is fitted to the scaled
observations in the identified region, and all stations within this region would share the same "regional" parameters. Station
specific parameters and quantiles can then be inferred by appropriate scaling. This method has been applied by various authors
(e.g. Gaál and Kyselý, 2009; Malekinezhad and Zare-Garizi, 2014) and on various distributions such as the GEV and the GP.
Variants of this method exist such as the region of influence (ROI) proposed by Burn (1990), that avoids defining fixed regions,
but assigns homogeneous region (neighborhood of different shapes according to the method) for each site. Scaled observations
within the neighborhood of each station is then used to estimate the regional parameters of that station. This method has been
applied by various authors (see Gaál et al., 2008; Kyselý et al., 2011; Carreau et al., 2013; Evin et al., 2016; Das, 2017, 2019).
In contrast to the aforementioned methods that generally rely on some covariates such as spatial coordinates to define the
homogeneous regions, another variant, recently developed by Gall et al. (2021) defines homogeneous regions based on the
similarity of their upper tail behavior. This method avoids the use of any covariate, but relies completely on the precipitation
data at hand. The upper tail behavior for each station is summarized based on a ratio of probability weighted moments (PWMs)
(refer to Eq 4). Subsequently a clustering algorithm is used to partition these ratios into distinct homogeneous regions, and
then regional parameters can be estimated.

   On the other hand, spatial methods exists in which all the observations from all the stations are pooled and then used to
estimate spatial surface for each of the model parameters. The surface for each of the model parameters is defined as a function
of some well chosen covariates such as longitude, latitude, altitude, etc. Estimating the parameters though involve simply
the estimation of the coefficients of these relationships. From the fitted surfaces, station specific model parameters can be





inferred as a function of the covariates at that specific location. Surfaces that are smooth and flexible can be obtained by fitting
generalized additive models (GAM) to the relationships (see Chavez-Demoulin and Davison, 2005; Blanchet and Lehning,
2010; Youngman, 2019, 2020). Other alternatives to the classical RFA includes the Bayesian spatial modeling (see Madsen
et al., 1995; Cooley et al., 2007) and those discussed in Cunnane (1988).

Recent analyses have been done to compare performance of regional approaches with particular interest in distributions
allowing to model extremes only. Gaál et al. (2008) compared different versions of the ROI method against the classical RFA
method of Hosking and Wallis (2005). The ROI versions were distinguished by the choice of the distance metric and the maxi-
mum threshold to delineate neighborhoods. For all the methods, GEV distribution was assumed as the underlying distribution.
The authors through a Monte Carlo simulation study concluded that the ROI approach was superior to the classical RFA in-
volving distinct clusters. In an interpolation framework, Carreau et al. (2013) compared three methods; spatial interpolation
of locally estimated parameters. method of ROI and a rainfall generator called SHYPRE. For the first two methods, GEV
distribution was assumed. The author found comparable performance between the ROI and SHYPRE, and lack of robustness
in the method based on interpolation of local parameters. Deidda et al. (2021) also using GEV compared the classical RFA of
Hosking and Wallis (2005) and geostatistical interpolation of locally estimated parameters. They highlighted the limitation of
the former in yielding distinct regions and being of less accuracy compared to the later. Other comparisons includes those of
Gaál and Kyselý (2009); Kyselý et al. (2011) and Das (2019).

Our approach differs from the aforementioned studies in the following aspects. First, in contrast to the case where the
underlying distributions are basically for modeling only extremes (e.g. Burn, 1990; Gaál et al., 2008; Gaál and Kyselý, 2009;
Kyselý et al., 2011; Carreau et al., 2013; Evin et al., 2016; Das, 2019; Deidda et al., 2021), we consider a model, the EGDP
(Naveau et al., 2016) that models both low, medium and extreme precipitations. This is inline with our goal of having a robust
and reliable model, that is able to model the whole distribution and not only the extremes. Secondly, our comparison approach
is more general and based on Garavaglia et al. (2011) and Renard et al. (2013), by focusing on the predictive ability of the
models in a cross validation framework similar to the case of authors such as Blanchet et al. (2015); Evin et al. (2016), rather
than simply based on quality of fit (e.g. Gaál et al., 2008; Kyselý et al., 2011; Deidda et al., 2021). Finally, in our contribution,
we compare new methods not previously compared viz-a viz. The first method defines distinct homogeneous regions based
on their similarity in upper tail behavior (Gall et al., 2021). The second method is based on the ROI approach framework of
Evin et al. (2016). The last method is a spatial approach that assumes generalized additive models (GAM) forms for the model
parameters. For all the methods, we assume the EGPD as the underlying marginal distribution. We apply this comparison on a
dense network of over 1100 daily stations located within Switzerland and in the neighboring countries.

The paper is organized as follows: Section 2 presents the data and the study area. Section 3 introduces the competing models
while section 4 describes the comparison methodology as well as the criteria used. The results are presented in section 5.
Finally we discuss the conclusion and the relevant perspectives in section 6.

## 2 Data and study area

The comparison is made by considering daily precipitation observations recorded at 1176 stations shown in the map of Figure 1. From this total, 500 are located within Switzerland and 676 in the neighboring countries. The data has variable length ranging
from a minimum of 20 years to a maximum of 156 years, within the period of 1863 to 2019. The bar plot in Figure 1 shows the number of stations installed in each country during each decade of the study period. The main study area is Switzerland, we use the data in the neighboring countries simply to improve the estimates of the stations located around the border of Switzerland. Consequently, though the whole stations (both within and outside) are used in the regionalization and model fitting, we apply the performance criteria only on the stations located in Switzerland.

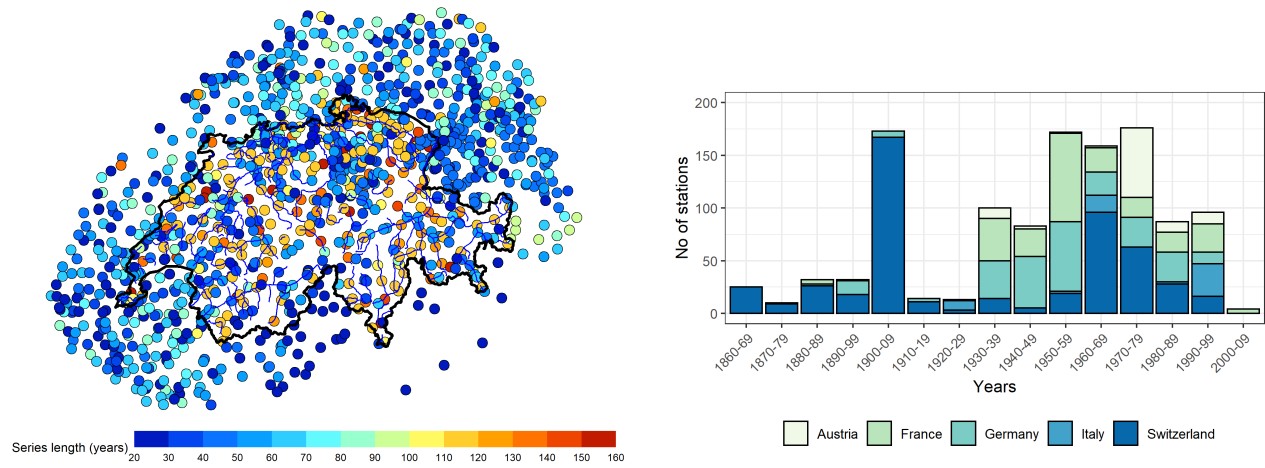

**Figure 1.** Description of the data used for the study. Left: Map of Switzerland and the neighborhood showing the location of the 1176 daily stations. The color indicates the length of the series, minimum of 20 years and maximum of 156 years. Right: Bar plot showing the number of stations installed in each country for each decade

Daily precipitation in Switzerland is characterized by seasonality arising from multiple moisture sources brought by prevailing winds (Sodemann and Zubler, 2009; Umbricht et al., 2013; Giannakaki and Martius, 2015). It is also marked by spatial variability both in intensity and occurrence resulting from the complex topography (Sevruk, 1997; Sevruk et al., 1998; Frei and Scha, 1998; Molnar and Burlando, 2008; Isotta et al., 2014). Winter receives the least precipitation. Summer is the main season of precipitation all over Switzerland, except for the Ticino, where autumn is the main season. The Ticino stands generally as the
region subjected to the heaviest precipitation. In the North of the country, the topographical effect is evident, with the northern rim and the Jura receiving heavier precipitation compared to the plateau.

Resulting from the marked seasonality of the precipitation, and the importance of taking this into account (Leonard et al., 2008; Garavaglia et al., 2011), necessitates a seasonal based analysis. We thus divide the data into the four distinct seasons of three months each: Winter (Dec, Jan, Feb), Spring (Mar, Apr, May), Summer (Jun, Jul, Aug) and Autumn (Sep, Oct, Nov). We
apply the comparison method on the seasons independently.





## 3 Candidate Methods

In this section, the various methods compared in the regionalization of the precipitation are described. First, we introduce the marginal distribution of the precipitation. Next we give an overview of the methods of regionalization that will be used in this paper. Finally, we summarize the regional models that will be compared.

### 3.1 Marginal distribution of positive rainfall

In this work, we use the marginal distribution of precipitation proposed by Naveau et al. (2016), which is able to model sufficiently the full spectrum of positive precipitation. The model is in compliance with EVT for both the upper and lower tail of the distribution, and provides a smooth transition in between. It though provides an alternative to the light-tailed distributions such as Gamma, that can underestimate extremes (Katz et al., 2002). Four parametric families of this model have been proposed by Naveau et al. (2016), and more recently a non-parametric scheme of the transition function by Tencaliec et al. (2020). However the simplest of the parametric family has been shown to be parsimonious and able to model adequately precipitation intensities, without the need for GPD threshold selection (Naveau et al., 2016; Evin et al., 2018; Gall et al., 2021). We thus use this model in our study.

Let $X$ be a random variable representing positive daily precipitation intensities that is distributed according to the EGPD, then the cumulative distribution function (CDF) is given by :

$$F(x) = \mathbb{P}(X \leq x) = G\left[H_\xi\left(\frac{x}{\sigma}\right)\right] \tag{1}$$

where G is any CDF that provides for a smooth transition between both upper and lower tail of the distribution, which are compliant with the EVT, and:

$$H_\xi\left(\frac{x}{\sigma}\right) = \begin{cases} 1 - \left(1 + \xi\frac{x}{\sigma}\right)_+^{-1/\xi} & \text{if } \xi \neq 0 \\ 1 - \exp\left(-x/\sigma\right) & \text{if } \xi = 0 \end{cases} \tag{2}$$

with $a_+ = \max(a, 0)$.

For the parsimonious model we use, the function G is simply defined as $G(v) = v^k$. Therefore the the model is given as:

$$F(x) = \left[H_\xi\left(\frac{x}{\sigma}\right)\right]^k \tag{3}$$

The model thus has three parameters. $k > 0$ controls the lower tail, $\xi \geq 0$ controls the upper tail, and $\sigma > 0$ is the scale parameter.

Inference of the model parameters can be done through maximum likelihood estimation (MLE), or through the method of probability weighted moments (PWM).





## 3.2 RFA based on upper tail behavior

Classical regional frequency analysis (Hosking and Wallis, 2005) defines regions that are homogeneous up to a scaling factor. To identify the regions, some covariates have to be carefully chosen, which normally includes some at-site characteristics such

as geographical and atmospheric characteristics. However this information might not be generally available at each station. Some homogeneity test then have to be applied to confirm that the regions are sufficiently similar.

Gall et al. (2021) proposed a fast and efficient method to delineate regions based on the homogeneity of their upper tail behavior. The method relies on the precipitation data at hand only without the need for additional covariates. More so, regions identified are inherently homogeneous, thereby avoiding the need for the application of some homogeneity tests. For each

station $i$, a ratio $\omega$ given in Eq. 4 that is based on probability weighted moments (PWM) is obtained.

$$\hat{\omega} = \frac{3\hat{\alpha}_2 - \hat{\alpha}_0}{2\hat{\alpha}_1 - \hat{\alpha}_0} - 1, \tag{4}$$

where $\hat{\alpha}_j$ denotes the PWM of order $j$.

The authors showed that $\omega$ summarizes the upper tail behavior of the data at hand, and for the EGPD model, it depends mainly on the $\xi$ parameter (effect of $k$ not very significant, (see Gall et al., 2021)). Stations with high values of $\omega$ have high

intense extremes, and those with low values have less intense extremes. The idea is then to classify or form regions that have similar value of $\omega$. This can be achieved by using any of the clustering algorithms such as K-means, hierarchical clustering, PAM, etc.

After forming plausible regions, the next step is to regionalize the estimate of the shape parameter. Given that the regions formed are based on the similarity of the intensity of their extremes (upper tail similarity), one way to proceed is to first get

the at-site estimate of the model parameters i.e $(k_i, \sigma_i, \text{and } \xi_i)$. For each homogeneous region, we then get the regional shape parameter $\xi_r$ as the average of all the local shape parameters in that region. Then finally, for each of the stations in that region, we impose $\xi_r$ and then get local $k_i$, $\sigma_i$ given this value of $\xi_r$. One can choose to do the fitting using maximum likelihood estimate (MLE), or the method of PWM. In our case however, we adopt the MLE estimate as with all the subsequent methods for the sake of comparison, since the method of PMW cannot be used with the GAM models.

To summarize, shape parameter regionalization using this method involves the following steps.

1. For each station $i$, $i = 1, \cdots, N$ use the positive data to estimate the ratio $\hat{\omega}_i$.

2. Decide on homogeneous regions using an appropriate clustering algorithm that is applied to $\hat{\omega}_1, \cdots, \hat{\omega}_i, \cdots, \hat{\omega}_N$ from the previous step.

3. For each homogenous region $C$,

(a) fit EGPD locally to find $(k_i, \sigma_i, \text{and } \xi_i)$,

    (b) find the regional shape parameter $\xi_r$ as average of all $\xi_i$ in that region,

    (c) fit EGPD locally again to find new estimates $\hat{k}_{i,new}$ and $\hat{\sigma}_{i,new}$, given the estimated $\xi_r$





We have also explored other options to estimate $\xi_r$ after obtaining the homogeneous regions:

- The first method involves pooling all the observations in a homogeneous region (cluster) after scaling them by their mean, and then fitting a regional EGPD on this scaled data to estimate the regional parameters $(k^{(R)}, \sigma^{(R)}, \xi^{(R)})$. We then retain $\xi^{(R)}$ and then refit an EGPD locally to estimate $k_i$ and $\sigma_i$. Every station in that cluster will have similar $\xi^{(R)}$ but locally estimated $k_i$ and $\sigma_i$.

- The second approach is similar to the main method where we take the the average of the locally estimated $\xi$, but here, we take a weighted average . Our idea is that, for each cluster, the locally estimated $\xi$ for the medoid station (the station with the least average dissimilarity to all the other stations in the same cluster) should be assigned the highest weight in the average, all other stations should then have weights as a function of their dissimilarity to this medoid. Thus stations that are very similar to the medoid should have higher weights, while those that are less similar should have smaller weights. The dissimilarity is measured by the Manhattan distance $|\omega_m - \omega_i|$, where $\omega$ is given in Eq. 4, while the indices $m$ and $i$ denote respectively the medoid and the station $i$.

After testing these three approaches to estimate $\xi_r$, by measuring the accuracy of the resulting quantile-quantile plot according to the normalized-root-mean-square-error (NRMSE) (see section 4 for details of this criteria), results (not shown here) showed that the first method, where we simply take the average of the locally estimated $\xi$, resulted in the least error. We thus retain this approach in our subsequent analysis.

## 3.3 RFA based on region of influence approach (ROI)

The region-of-influence (ROI) method (Burn, 1990) is similar in concept to the classical RFA method. It circumvents the drawback of having contiguous regions separated by distinct boundaries that results in "undesirable step changes of the variables and estimated quantiles" (Gaál et al., 2008). Instead of defining distinct homogeneous regions separated by some boundaries, a region of influence is assigned to each station. All the scaled observations in the identified ROI for that station are used to estimate its regional parameters. To apply this method, several choices have to be made. These involve the choice of the scale factor, distance metric, radius delimitation and homogeneity test. The choices have influence on the application of the method and have to be carefully and objectively decided. Different authors in the application of the methods have explored some or all of these factors, starting from Burn (1990), and in Gaál et al. (2008).

In this work, we follow the objectively selected steps and choices similar to Evin et al. (2016) in the application of the method in the Southeastern part of France. The authors applied the method by considering POT (exceedances of a 70 % quantile) of central rainfalls (largest observations in 3 day rainfall events) and on some distributions (Exponential, GPD and Weibull). We apply the same procedure but on positive rainfall and on EGPD model.

Let $X_i \sim EGPD(k_i, \sigma_i, \xi_i)$ be the random variable of daily positive rainfall at station $i$ that is distributed according to the EGPD. We assume also that $Y_i = \frac{X_i}{m_i}$ is the daily positive rainfall normalized by a scale factor $m_i$. Then, if we consider several stations (whose data has been normalized as well) that have similar distribution as $Y_i$ and use the data to estimate the regional parameters $k^{(R)}, \sigma^{(R)}$, and $\xi^{(R)}$ , then, $Y_i$ will have parameters $(k^{(R)}, \sigma^{(R)}\xi^{(R)})$. Accordingly, by back transformation, the





unnormalized random variable $X_i$ will have the parameters $(k^{(R)}, m_i\sigma^{(R)}, \xi^{(R)})$. This shows that for a random variable that is distributed according to the EGPD, after regionalization, the parameters $k$ and $\xi$ are those obtained regionally, while the scale parameter $\sigma$ has to be multiplied by the scale factor for that station.

Using this method, we test two regional versions of the EGPD model.

1. The full regional model, where $X_i \sim EGPD(k^{(R)}, m_i\sigma^{(R)}, \xi^{(R)})$. We call this model ROI_EGPD_Full afterwards.

2. The semi regional model, where $X_i \sim EGPD(k_i, \sigma_i, \xi^{(R)})$: In this case, we retain the regional shape parameter $\xi^{(R)}$ obtained from the neighborhood, and estimate the two other parameters locally from only the data at the station $i$. We refer this model as ROI_EGPD_Semi.

The general procedure of application is summarized below and the details can be found in Evin et al. (2016).

1. For each station and season, exceedances of threshold of $95\%$ quantile (POT) are selected and scaled by a factor. The scale factor is the mean of all positive daily precipitation.

2. We start the search from a radius of 2 [km] starting from the current station. If other stations are found within this radius, we apply a homogeneity test on the scaled POT found in step 1. If the test is positive, we increase the radius by another 2 [km] and repeat the test. We stop the search when the test fails or when we reach a maximum radius beyond which we doubt the existence of homogeneity. We use 100 [km] as the upper bound.

3. The distance metric used is the "crossing distance" (Gottardi et al., 2012) given in Equation 5. This distance takes into account the effect of elevation and is summed over all the pixels along a straight line between two targeted stations. We use a weight on elevation equal 20 similar to Evin et al. (2016) to account for the effect of relief. Again, following the same authors, we use the test of Hosking and Wallis (2005) on mean and L-coefficient of variation (L-CV).

$$d = \sqrt{\sum_{pixels} \left(\Delta x^2 + \Delta y^2 + 20\Delta z^2\right)} \tag{5}$$

4. For all stations within the ROI of the current station, we estimate the regional parameters by MLE on the scaled positive observations We also apply weights in the likelihood similar to Evin et al. (2016) that penalize the observations of the stations in the ROI depending on their proximity in distance to the target station. We also allow for censored MLE to reduce the effect of observations close to zero.

## 3.4 Spatial method based on Generalized Additive Model (GAM)

In contrast to the previous methods in which regionalization is based on homogeneity of normalized data or upper tail similarity, this is a regression-based method for fitting the parameters of a model, by allowing for spatial non-stationarity of the model parameters. We thus pool all the observations from all the stations to estimate flexible and smooth spatial surface for each parameter, relying on the ground that pooling of spatial information can help improve the at-site estimates, and hence the extreme quantiles. In particular, we let the parameters to have a generalized additive model (GAM) form, represented by





smoothing splines. In effect, we assumed them to have some form of flexible relationship with some covariates $x$, that can be explained by GAM forms.

For the EGPD model, we have $X(\boldsymbol{x}) \sim EGPD(\sigma(\boldsymbol{x}), k(\boldsymbol{x}), \xi(\boldsymbol{x}))$, where $\boldsymbol{x}$ denotes some covariate, and each of the model parameter depends on some form of $\boldsymbol{x}$. The relationship between the model parameter (say $\alpha$) and the covariate $\boldsymbol{x}$ is through an identity link:

$$\alpha_{(\boldsymbol{x})} = \beta_0 + \sum_{k=1}^{K}\sum_{d=1}^{D_k} \beta_{kd} b_{kd}(\boldsymbol{x}), \tag{6}$$

where $\beta_{kd}$ and $b_{kd}$ are the basis coefficients and the basis functions respectively. $K$ is the number of smooths and $D_k$ is the dimension (number of knots) for smooth $k$.

For the choice of spatial covariates, we use longitude, latitude and mean daily precipitation because they give better Akaike information criterion (AIC) (Akaike, 1974). To fit EGPD with GAM, we extended the functions already available in the `evgam` R package (Youngman, 2020)

## 3.5 Summary of the models that are compared

This section gives an overall summary of the regional models that are compared in the present study. Table 1 presents the four regional models plus the local EGPD model. The four models fall under three categories: Regional model involving "hard" clustering (No 2 in the Table). Regional models based on ROI (No 3 and 4). Spatial models based on GAM (No 5). For details of the three classes, refer to Sections 3.2, 3.3 and 3.4 respectively.

**Table 1.** Summary of the regional models that are compared in this study. The first model is the local EGPD model, the next three models are based on regional homogeneity, while the last model is a spatial methods based on GAM. The second column gives the name of the model. The last three columns are the parameters of the EGPD model, and indicates whether the parameter is estimated locally (from the data of the station at hand only) or though regionalization.

| Model | Methods | $\kappa$ | $\sigma$ | $\xi$ |
|---|---|---|---|---|
| 1 | Local EGPD | local | local | local |
| 2 | Omega EGPD | local | local | regional |
| 3 | ROI EGPD Full | neighborhood | neighborhood | neighborhood |
| 4 | ROI EGPD Semi | local | local | neighborhood |
| 5 | GAM EGPD | spatial | spatial | spatial |

The models are distinguished by the following:

1. Model 1 is the local EGPD model, without any regionalization.

2. In the case of Omega_EGPD, we regionalize only the $\xi$ parameter of the EGPD model according to RFA based on upper tail behavior. We then estimate $\kappa$ and $\sigma$ locally, at each station.





3. For Model 3, ROI_EGPD_Full, we regionalize all the three parameters based on ROI approach.

4. Model 4, ROI_EGPD_Semi is a semi regional version of model 3. We retain the regional estimate of $\xi$, and then estimate $\kappa$ and $\sigma$ locally, for each station.

5. Finally GAM_EGPD is a spatial model based on GAM, we fit an EGPD model directly and obtain the three EGPD parameters from the fitted smooth surfaces.

## 4 Comparison method and evaluation criteria

This section introduces the method of comparison as well as the performance criteria used to make regional comparison of the candidate methods.

The evaluation framework and criteria is as proposed by Garavaglia et al. (2011); Renard et al. (2013). Garavaglia et al. (2011) and the references there in, argued that the classical statistical goodness of test fits such as Kolmogorov– Smirnov test (Kolmogoroff, 1941; Sminorv, 1944) , Anderson– Darling Test (Anderson and Darling, 1952) , Cramer von-Mises criterion (Cramer, 1928; Darling, 1957) lack the ability to assess the models ability to predict unobserved values, and that they are also not very efficient for three-parameter distributions.

Accordingly, we follow a split sampling procedure and a cross-validation framework. For each station $i$, we used $1/3^{rd}$ of the data by choosing every third observation to reduce temporal dependence. Then, we divide the non-zero observations into two equal sub-samples of the same length but on different years that are randomly chosen. We call the first and second sub-samples $S_i^{(1)}$ and $S_i^{(2)}$. We then fit a model $\hat{F}_i^{(1)}$ and $\hat{F}_i^{(2)}$ on sub-sample $S_i^{(1)}$ and $S_i^{(2)}$ respectively. We then compute the criteria $C_i^{(12)}$ at station $i$, by comparing $\hat{F}_i^{(1)}$ vs $S_i^{(2)}$ (i.e model fitted on sub-sample 1 vs observations in sub-sample 2). In the same way we compute criteria $C_i^{(21)}$. Given that we have $N$ stations, and so $N$ values of both $C_i^{(12)}$ and $C_i^{(21)}$, the regional score is obtained as the average of these scores. This procedure is repeated 50 times to obtain 50 regional averages of these indices.

For each method, four (4) criteria $C$ are computed. We first judge the methods based on how they accurately fit the entire observations at each site. Next, we compare them in terms of their robustness in extrapolation, *i.e.* how stable a high quantile estimate is, depending on which sub-sample is used in the estimate. We finally judge the performance based on the reliability to predict rainfall maxima.

In the following paragraphs, we describe the four criteria used for the comparison:

### 4.1 Accuracy of the whole distribution

The accuracy of the model in predicting the positive observations at a given station is given by the normalized root mean square error (NRMSE) (Blanchet et al., 2019). For each site $i$, the positive observed values in $S_i^{(2)}$ are associated with their empirical return periods. We then use the fitted model on $S_i^{(1)}$, i.e $\hat{F}_i^{(1)}$, to estimate the modelled quantiles associated to these return





levels and finally compute the root mean squared errors associated to these quantiles which is normalized by the average daily rainfall. This score is given as:

$$
\text{NRMSE}_i^{(12)} = \frac{\left\{ \frac{1}{n_i^{(2)}} \sum_{k=1}^{n_i^{(2)}} \left( r_{i,T_k}^{(2)} - \hat{r}_{i,T_k}^{(1)} \right)^2 \right\}^{1/2}}{\frac{1}{n_i^{(2)}} \sum_{k=1}^{n_i^{(2)}} r_{i,T_k}^{(2)}}
\tag{7}
$$

where $\text{NRMSE}_i^{12}$ is the score computed at station $i$, $r_{i,T_k}^{(2)}$ is the $k^{th}$ observation of return period $T$ in $S_i^{(2)}$, $\hat{r}_{i,T_k}^{(1)}$ is the

285 corresponding $T$ return level estimated from $\hat{F}_i^{(1)}$. The denominator $\frac{1}{n_i^{(2)}} \sum_{k=1}^{n_i^{(2)}} r_{i,T_k}^{(2)}$ is the average daily rainfall at site $i$. Details on the score is given in Blanchet et al. (2019).

Finally, the regional score computed over the $N$ stations, *i.e.* $\text{NRMSE}_{\text{reg}}^{(12)}$ is given as:

$$
\text{NRMSE}_{\text{reg}}^{(12)} = 1 - \frac{1}{N} \sum_{i=1}^{N} \text{NRMSE}_i^{(12)}.
\tag{8}
$$

$\text{NRMSE}_{\text{reg}}^{(21)}$ is computed in similar way. We thus finally have $2 \times 50$ values of $\text{NRMSE}_{\text{reg}}$ resulting from the cross-

290 validation on both periods. $\text{NRMSE}_{\text{reg}} = 1$ means a perfect model, and the closer the value is to 1, the more accurate the model is.

## 4.2 CRPS

The continuous ranked probability score (CRPS) has been used as a metric to compare the performance of two competing probabilistic forecasts models (Jordan et al., 2018) . It gives a combined measure of both spread and reliability of a forecast

distribution, given an observation or outcome that has been observed.

For a given observation $x_{i,t}$ at station $i$ and time step $t$ that is contained in $S_i^{(1)}$, we have 50 of its quantile estimates coming from the 50 fitted models $F_i^{(2)}$ ( models fitted with data not containing $x_{i,t}$ ). If the method used to estimate all this 50 models is accurate enough, then these 50 quantile estimates should be similar (low spread), and very close to the observed value $x_{i,t}$. The same applies to an observation $x_{i,t}$ contained in $S_i^{(2)}$ when compared to its quantile estimates from the 50 models of $F_i^{(1)}$.

Thus the CRPS of $x_{i,t}$ should be low, and when applied to all the observations at station $i$, the average, $\overline{\text{CRPS}_i}$ should be also low.

The $\overline{\text{CRPS}_i}$ averaged over the observed data from time step $t = 1$ to $t = T_i$ at station $i$ is given as:

$$
\overline{\text{CRPS}_i} = \frac{1}{T_i} \sum_{t=1}^{T_i} \int_{\mathbb{R}} \left\{ F_{i,t}(y) - H(y - x_{i,t}) \right\}^2 \, \mathrm{d}y,
\tag{9}
$$

where $H(z)$ denotes the Heaviside function that is 0 if $z \leq 0$ and 1 otherwise. $F_{i,t}(y)$ and $x_{i,t}$ are the CDF of the 50

estimates, and the observed value at time step $t$ of station $i$ respectively. Note that for the cross-validation, if $x_{i,t}$ belongs to $S_i^{(1)}$ (resp. $S_i^{(2)}$), then $F_{i,t}(y)$ will be the CDF of the 50 quantile estimates of the 50 models $F_i^{(2)}$ (resp. $F_i^{(1)}$). The smaller the $\overline{\text{CRPS}}$ score, the better the model.





Giving that we have $N$ stations, at the end, we will have $N$ values of $\overline{\mathrm{CRPS}}$ computed for each of the competing model. This is in contrast to the other criteria for which we have either 50 or 100 values per model.

### 4.3   Stability of high quantile estimate

The robustness of a method is determined by the stability of a high quantile estimate form one sub sample to another. A robust method should thus have similar estimate of say, 100 year event, when the subsample/calibration data is changed. The SPAN criteria (Garavaglia et al., 2011; Blanchet et al., 2019) gives the measure of the stability of a chosen quantile estimated from two sub-samples.

The score is computed as the absolute difference between the two quantile estimates divided by their average and is given as:

$$\mathrm{SPAN}_{i,T} = \frac{2 \left| \hat{r}_{i,T}^{(1)} - \hat{r}_{i,T}^{(2)} \right|}{\left( \hat{r}_{i,T}^{(1)} + \hat{r}_{i,T}^{(2)} \right)} \tag{10}$$

where $\hat{r}_{i,T}^{(1)}$ and $\hat{r}_{i,T}^{(2)}$ are the $T$-year return levels estimated from $\hat{F}_i^{(1)}$ and $\hat{F}_i^{(2)}$ respectively at station $i$.

The score is computed for all the $N$ stations and the regional score $\mathrm{SPAN}_{\mathrm{reg},T}$ is computed as:

$$\mathrm{SPAN}_{\mathrm{reg},T} = 1 - \frac{1}{N} \sum_{i=1}^{N} \mathrm{SPAN}_{i,T} \tag{11}$$

At the end we have 50 values of SPAN and a robust model should have a $\mathrm{SPAN}_{\mathrm{reg},T}$ of 1, therefore the closer the value is to 1, the more stable/robust the model is.

### 4.4   Reliability in predicting the maximum observed value

The reliability of a model is defined as its ability to associate the correct probability to a given observation. Specifically, the $FF$ criteria measures the reliability of the model in predicting the maximum value in a given sample. The score is defined as:

$$FF_i^{(12)} = \left[ \hat{F}_i^{(1)} \left( \max_i^{(2)} \right) \right]^{n_i^{(2)}} \tag{12}$$

where $FF_i^{(12)}$ is the cross validation criteria computed at station $i$, by predicting the probability of the maximum value in sub-sample 2, $S_i^{(2)}$, of sample size $n_i^{(2)}$ using the model $\hat{F}_i^{(1)}$ fitted on the sub-sample 1, $S_i^{(1)}$.

According to Renard et al. (2013) and Blanchet et al. (2015), if the model is reliable, then $FF_i^{(12)}$ is realization of a uniform distribution. Accordingly, if we compute the score for all the stations, *i.e.* $FF_1^{(12)}, \cdots, FF_i^{(12)}, \cdots, FF_N^{(12)}$, we should end up with a set $FF^{(12)}$ of $N$ realizations of a uniform distribution. Blanchet et al. (2015) therefore, concluded that, the area between the density of $FF^{(12)}$ and an uniform density should be close to zero.



The regional score $FF_{reg}^{(12)}$ is computed as $1 - \text{AREA}(FF^{(12)})$ and $FF_{reg}^{(21)}$ is computed in similar way. The closer the value is to 1, the more reliable the model is in prediction of the maxima. We have at the end $2 \times 50$ values of $FF_{reg}$ .

## 5 Results

### 5.1 Estimated regions with RFA by upper tail behavior

We follow the steps of this method application as itemized in section 3.2. We proceed on a seasonal basis. We first divide the data into four seasons and then estimate $\hat{\omega}_i$ for each station.

The next task is to define the regions that are homogeneous, or have similar value of the ratio $\hat{\omega}$. Two choices are necessary; the clustering algorithm, and the optimal number of clusters. Gall et al. (2021) in application of this method to Switzerland daily precipitation data simply used the partitioning around medoids (PAM) algorithm alongside the Silhouette criteria (Rousseeuw, 1987) to determine optimal cluster numbers. We investigated two other clustering algorithms; K-means, and hierarchical agglomerative clustering (HAC). In the case of HAC, we also explored its various linkage criteria. For details of these clustering methods see Kaufman and Rousseeuw (2005).

To determine the optimal number of clusters, we computed some internal validation criteria. These criteria evaluates the quality of the resulting clusters by relying on some properties of the cluster, without resorting to some external information.

We computed a couple of these internal validation criteria in addition to Silhouette (Rousseeuw, 1987) such as the criteria by Dunn (Dunn, 1974), Davies Bouldin (DB) (Davies and Bouldin, 1979), Xie Beni (Xie and Beni, 1991), and S_Dbw (Halkidi and Vazirgiannis, 2001). For the details of these criteria and their formulations, refer to the associated references.

To decide on the clustering algorithm and the internal validation criteria, we performed a simulation study similar to Brun et al. (2007), by checking the misspecification rate of each of the criteria, given a known prior partition. Finally (result not shown), PAM emerged as the most appropriate algorithm, alongside Silhouette, DB (Davies and Bouldin, 1979), and S_Dbw (Halkidi and Vazirgiannis, 2001).

Subsequently, for each season and station, we compute $\hat{\omega}_i$, we use the PAM algorithm to partition the data into $K$ number of clusters, for $K = 2, \cdots, 10$. We finally retain the $K$ optimized by the majority out of the three criteria, i.e., Silhouette, DB and S_Dbw.

Fig 2 shows the optimal number of clusters identified for each season. We have 3 clusters in the case of winter (DJF), 2 in spring (MAM), 3 clusters in summer (JJA) and 2 in autumn (SON). Notably, although the spatial coordinates are not used, the identified clusters are somehow spatially plausible for each season. Stations in the South are generally in the same cluster. In the North, the stations located in the Northern rim and the Jura generally falls in the same cluster. This is partly according to our knowledge of the spatial pattern of heavy precipitations in the respective seasons. A few stations for each season however appeared in clusters different from their neighbors.





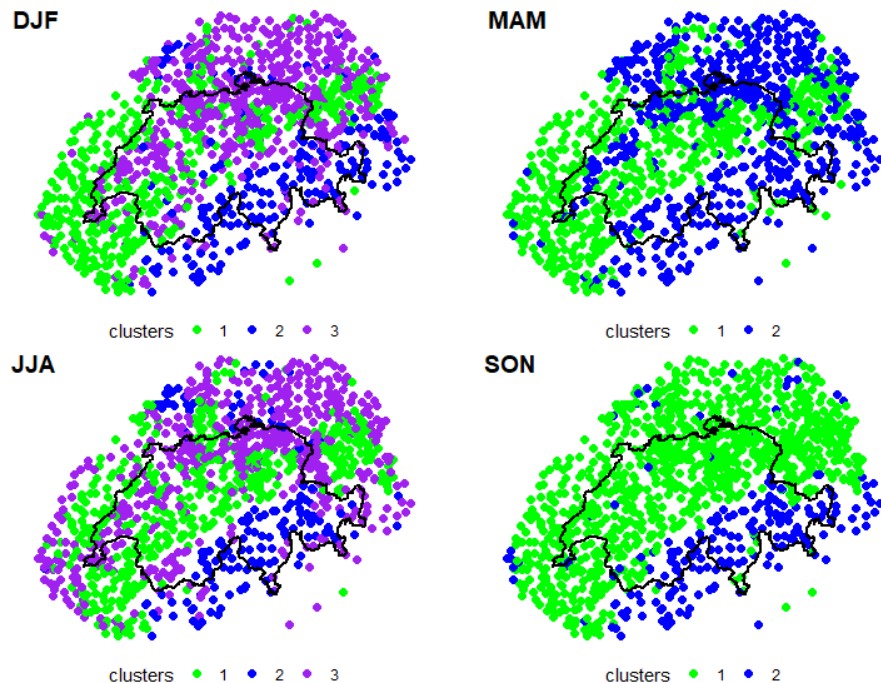

**Figure 2.** Maps of Switzerland showing the optimal number of clusters identified with the PAM algorithm for each season. For each season, the regions identified are color coded. From top left, going clock wise, DJF (2 clusters), MAM (2 clusters), JJA (3 clusters) and SON (3 clusters)

## 5.2 Estimated Regions with method of ROI

Figure 3 presents two maps describing the neighborhoods found for each station. On the left is the map of Switzerland showing the stations. The circle size corresponds to the size of the radius identified, while the color indicates whether finally a local fit will be done (for stations without any neighbors), or a regional fit (for stations with at least one neighbor). For those without neighbors, the search terminated sometimes at a very small radius indicating that although there are proximate stations according to the crossing distance, but after subjecting them to homogeneity test, it failed. For some stations however, no neighbors are found. This type of stations are those whose closest neighbors are located at a large crossing distance, and the test failed after application. On the right of Figure 3 the histogram of the distribution of the neighborhood size is shown. Only few stations reach the bound of 100 km.

In our subsequent experiments, we keep these neighborhoods and fit accordingly the model versions described in section 3.3, *i.e.* ROI_EGPD_Full and ROI_EGPD_Semi. For all the stations without any neighborhood, we simply fit a local EGPD model.





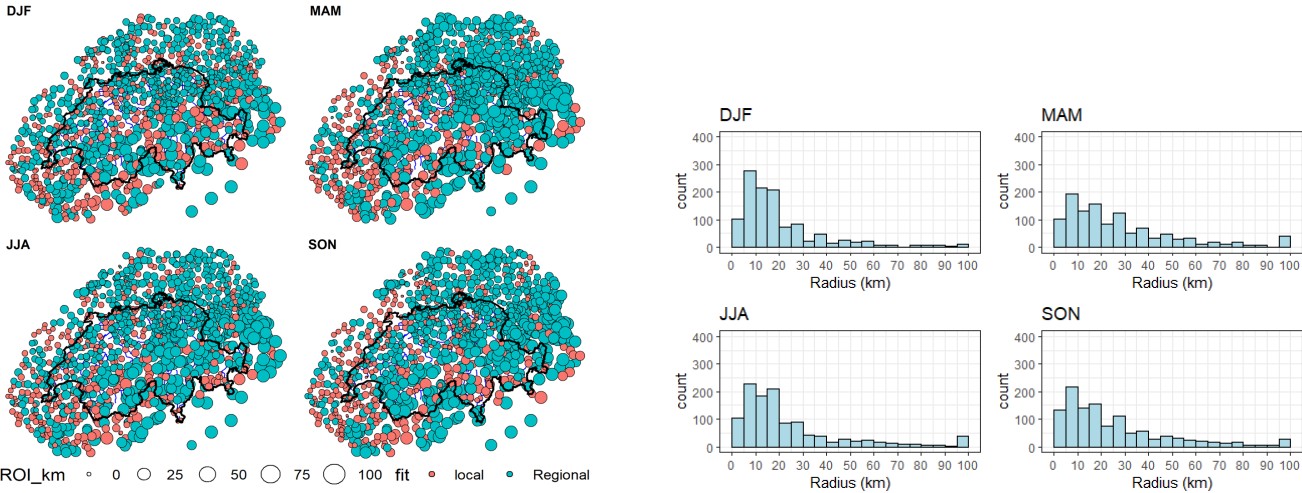

**Figure 3.** Properties of the radius of neighborhoods identified for each station. Left: Seasonal maps showing the size of the radius per station. For each season, the size of the circle is proportional to the radius size. the smallest is 0 [km], and the largest is 100 [km]. The color of the circle indicates whether a local fit is done, or a regional fit. Right: Histogram of the size of the radius identified.

## 5.3   Choice of covariates in GAM

Following the outlined methodology for the spatial model in section 3.4, we present in this part the choice of covariate combinations made for the EGPD model.

We use longitude, latitude and the mean daily precipitation to explain the parameters of EGPD, that is $k$, $\sigma$ and $\xi$. Other covariates would be possible but we use these ones because they are readily available and so estimation at ungauged locations would be possible. After testing different combinations of the covariates, we use the following forms for the relationships:

$$k = s(m) \tag{13}$$
$$\sigma = s(lon, lat) + s(m) \tag{14}$$
$$\xi = s(lon, lat) \tag{15}$$

where $s(lon, lat)$ means a thin plate spline smooth $s$ on the longitude $lon$ and latitude $lat$, $s(m)$ means a cubic spline on the mean daily precipitation $(m)$.

Although all the three model parameters have to be positive, we still used identity link function to reduce the complexity in generation of the gradients of the negative likelihood function of the EGPD model (necessary for model fitting in GAM, see Wood et al., 2016). We however imposed constraints in the likelihood function to ensure that the parameters remain positive.





## 5.4    Model comparison

In this section, we present the results of the comparison between the competing models, as judged by the criteria introduced in Section 4. We remind that the sampling was repeated 50 times for all the models, and so for clarity, we show the boxplots of the criteria. Each boxplot contains 50 values of the criteria obtained per run in the case of SPAN, 100 in the case of FF and NRMSE, and 500 in the case of CRPS (500 corresponds to the number of stations in Switzerland, for which the criteria was computed on).

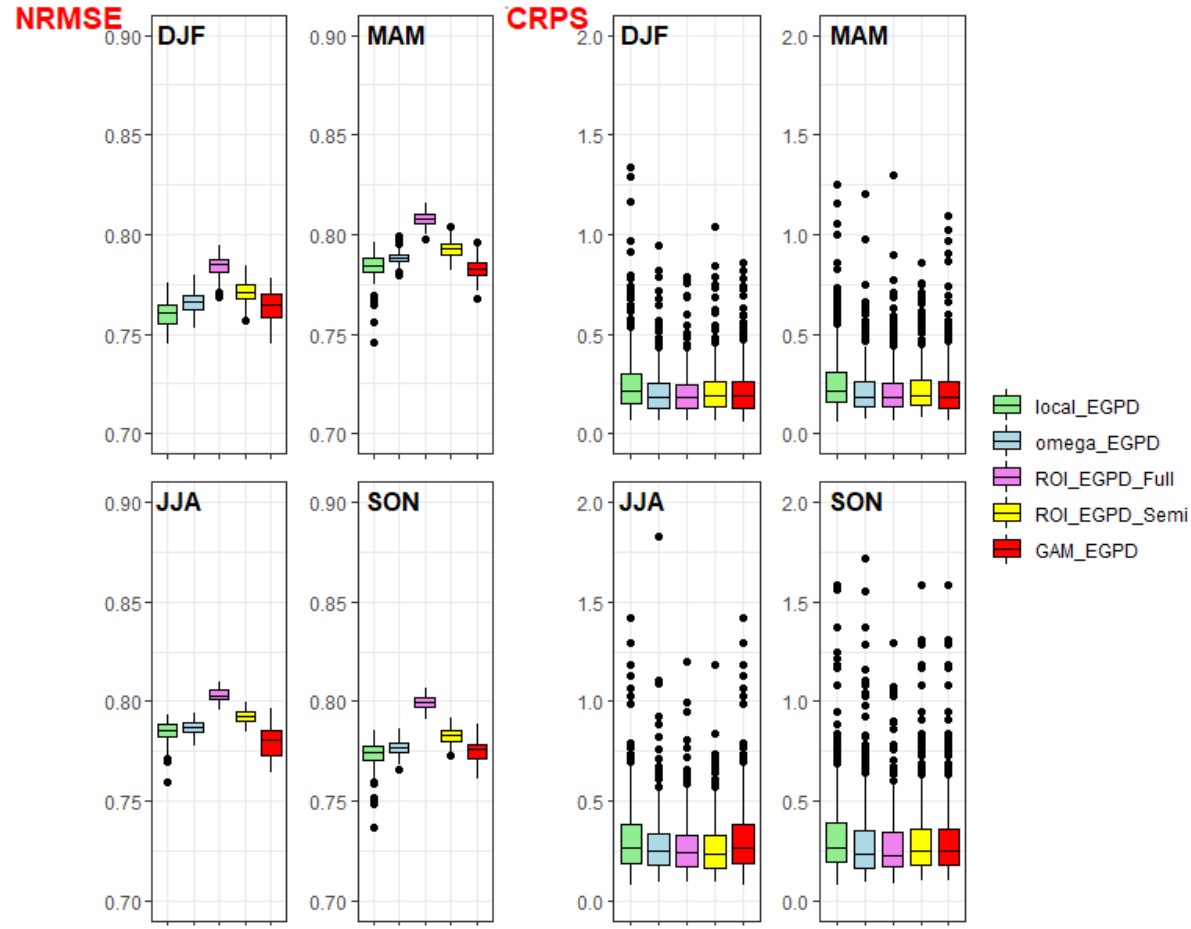

**Figure 4.** Criteria applied on the bulk of the distribution for each season. Left: Accuracy of the whole distribution as measured by the NRMSE, each boxplot contains 100 values. Right: CRPS score, each boxplot contain 500 points, 1 per station

First, the accuracy/reliability of the models in the bulk of the distribution as measured by the NRMSE is shown on the left of Figure 4. The results are shown per season and the closer the value is to 1, the more accurate the model is. From this result we can clearly see that for all seasons, the two models based on ROI are the most reliable. ROI_EGPD_Full model



(where we regionalize according to the method of ROI, all the three parameters of the EGPD model) being the best model compared to ROI_EGPD_Semi model 4 (where only the shape parameter is regionalized, but the other two parameters are locally estimated). The performance of the other regional models is similar and there is no large improvement of these models over the local EGPD model according to this criteria.

The CRPS score gives a combined measure of the spread and reliability of the competing models. A model should not only assign the correct probabilities to the observations, but the spread of the probabilities estimated from the different sub samples (100 in our case) should be low as well. We computed this score for all the positive observations at every station. The best model should have the smallest score for this model.

The plot on the right of Figure 4 presents the seasonal boxplots for the CRPS of the 5 models. Each box plot consist of 500 points, 1 per station (Recall the scores are computed for only the 500 stations located within Switzerland). From the boxplots and the medians, it is clear that ROI_EGPD_Full has the smallest value of this score. For all seasons, (except summer) the local model has the largest score, showing that the models offer improvement over the local model.

The stability/robustness of the estimate of a 100-year return level in between two periods as measured by the SPAN criteria is shown on the left of Figure 5. Obviously all the regional models show clear improvement in robustness over the local EGPD model. The spatial model (GAM_EGPD) shows the highest robustness over all the models, except in autumn when it is slightly overtaken by ROI_EGPD_Full model . The results also shows that of all the four regional models, the RFA model based on upper tail (Omega_EGPD) has the smallest robustness. In the case of the two ROI models, ROI_EGPD_Full is more robust compared to the ROI_EGPD_Semi model. The former involves regionalizing all the model three parameters while the later involves only the shape parameter regionalization. Looking at the two models where only the shape parameter is regionalized, the model based on ROI (ROI_EGPD_Semi) is more robust in all the seasons compared to the model based on upper tail behavior that involves clustering (Omega_EGPD).

Finally, the FF score measures the reliability of the models in the upper tail, more precisely in the prediction of the maximum observed value. This criteria is also optimized at a value of 1. The plot on the left of 5 shows generally high values of this score for all the models, indicating that they are generally reliable in the prediction of the maximum observed value. For all seasons all the regional models appear to be more reliable compared to the local model. An exception is however in the case of the model (Omega_EGPD) in winter and ROI_EGPD_Semi in summer (looking at the median). In autumn and summer, GAM_EGPD model emerges as the most reliable. Whereas in winter ROI_EGPD_Full is the most reliable, ROI_EGPD_Semi is the most reliable in spring.

To conclude, we present an overall summary of the results in Table 2 by comparing the median of the boxplots from the four criteria. We also show on Figure 6 map of Switzerland showing 100year return level of daily precipitation predicted with the ROI_EGPD_Full model for the four seasons. The maps reveals clear seasonality and spatial pattern. Ticino in the south is subjected to the highest levels in comparison to the other regions. This is for all seasons, but especially in the case of autumn where up to 400mm can be expected. Winter has the least in comparison to the other seasons.





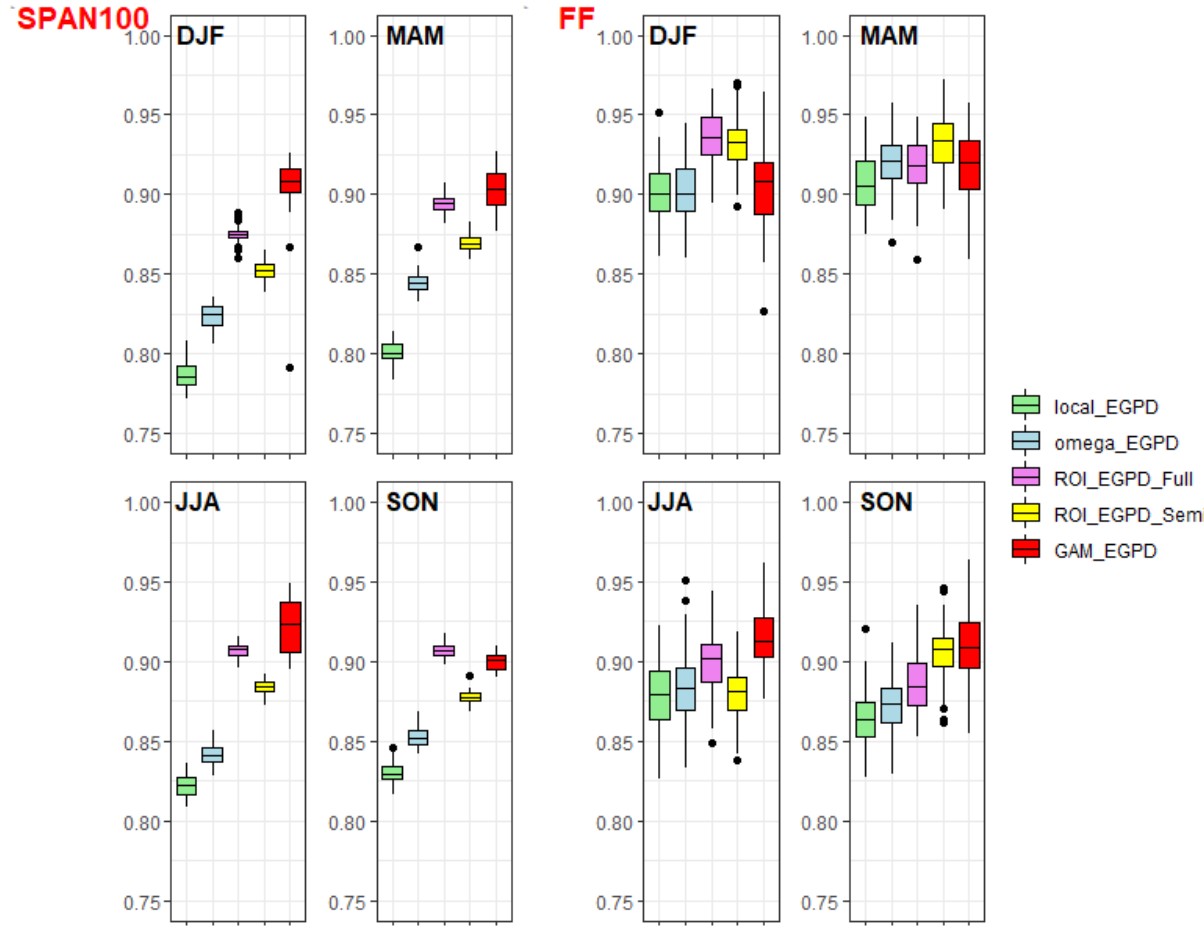

**Figure 5.** Criteria applied on the upper tail for each season. Left: Robustness of the local EGPD and the four candidate models, as measured by the SPAN criteria. The stability is measured with respect to a 100-year return level estimate. Each boxplot contain 50 values. Right: Reliability in prediction of the maxima as measured by the FF criteria, each boxplot contain 100 values.

**Table 2.** Summary of the comparison results from the four criteria used. For each season and criteria, the model with the highest median is shown. In the case of the CRPS score however, the model with the smallest median CRPS is shown.

| Season | NRMSE | CRPS | SPAN 100 | FF |
|--------|-------|------|----------|-----|
| Winter | ROI_EGPD_Full | ROI_EGPD_Full | GAM_EGPD | ROI_EGPD_Full |
| Spring | ROI_EGPD_Full | ROI_EGPD_Full | GAM_EGPD | ROI_EGPD_Semi |
| Summer | ROI_EGPD_Full | ROI_EGPD_Semi | GAM_EGPD | GAM_EGPD |
| Autumn | ROI_EGPD_Full | ROI_EGPD_Full | ROI_EGPD_Full | GAM_EGPD |





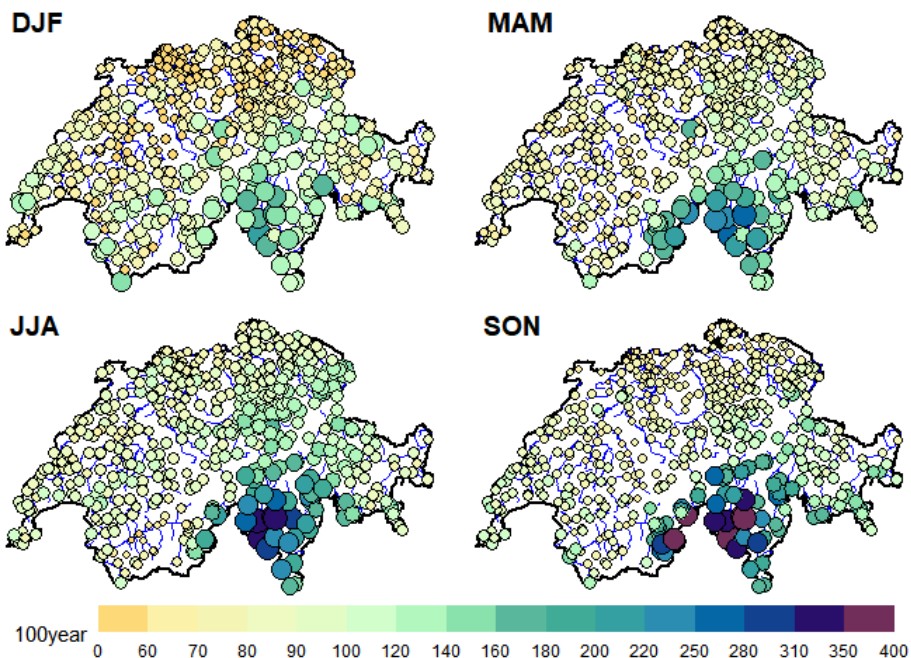

**Figure 6.** Map of Switzerland showing the 100year return level for the four seasons as predicted with ROI_EGPD_Full model.

## 6 Conclusions and Discussion

The objective of the paper was to objectively compare methods to improve the at-site estimates of daily precipitation. By considering a dense network of 1176 stations mainly located in Switzerland, we compared four methods based on different philosophies to regionalize the estimate of the daily precipitation extremes. We started with methods that are based on the philosophy of "regional homogeneity" where regions are identified based on the statistical homogeneity of the normalized data. The data is then pooled to estimate the parameters of the region. From this class, we consider a method, simple and efficient that uses only the precipitation data at hand to classify stations into regions that are homogeneous in their upper tail behavior. On the other hand, another method that avoids defining "hard" regions, but assumes that every station has its own homogeneous region that can be identified, subject to application of homogeneity tests. The other methods are spatially based, in which all the data is used to estimate smooth flexible surfaces for the parameters of the model. Pooling the data to estimate the surfaces thus ensures sharing of information between stations. We based our comparison by computing some criteria that measures the accuracy, robustness and reliability of the models in a cross-validation framework. More specifically, we assessed both the bulk of the distribution (NRMSE and CRPS), and the upper tail (SPAN and FF).

In contrast to most comparative studies of regionalization approaches where extreme distributions (GEV or POT ) were considered, we assumed the daily data to follow an Extended Generalized Pareto distribution (EGPD), able to model adequately the full spectrum of precipitation intensities. In this model both the upper and lower tails are compliant with the EVT and it offers a smooth transition for the bulk of the distribution.





The results showed that regionalization offered improvement in robustness and reliability even in the case of a full-scale model (EGPD) that includes the whole data in the estimate of its parameters.

From the four sets of criteria used, the performance depends on the season. Generally however, the following can be concluded:

- In terms of the reliability/accuracy over the whole distribution, the ROI model (ROI_EGPD_Full) where all the EGPD parameters were regionalized emerged as the most accurate.

- Reliability in prediction of the maxima as measured by FF indicated that the GAM model is the most reliable, especially
the seasons with the heaviest rainfall (summer and autumn).

- GAM model on the EGPD emerged as the most robust, and is followed closely by the the ROI model (ROI_EGPD_Full).

In conclusion, two models compete hand in hand, the ROI model (ROI_EGPD_Full) and the GAM model. Looking at the bulk of the distribution (NRMSE and CRPS), ROI_EGPD_Full is the best model. Focusing on the far tail however (FF and SPAN), GAM model is the best. As Garavaglia et al. (2011) pointed out, the two properties of reliability and robustness are
complementary. For two models of similar reliability, the model with the best robustness should be preferred. Given this, GAM model on EGPD, combining both properties in the upper tail can be said to be the preferred method. We note however, a major drawback of GAM. It requires significant computational time as compared to the ROI, especially in our case where we have a dense network with long series (up to 156 years for some stations) and we use all positive precipitation. In practise thus, it would be much easier to use ROI compared to GAM, given that the performances of both is similar, and the former is more
reliable when we consider the entire of the distribution (a feature of interest in our case), not only maxima.

Its worthy to note that in the course of the present study, we focused our evaluation at the station level, where we have observations. A further step will be to assess the models more generally by looking at their performances at ungauged locations in spatial cross-validation as done by Blanchet and Lehning (2010), Carreau et al. (2013) or more generally the framework proposed by Blanchet et al. (2019). In this aspect, the spatial model based on GAM, offers a key advantage over the other
methods since it inherently results in a regional model that can be applied everywhere. In the case of the other methods however, the step of choosing the appropriate interpolation technique has to be considered. Of worthy of mentioning also is the inherent drawback of conventional RFA approaches involving "hard" clustering, in this case the Omega_EGPD model. They are known to produce abrupt parameter shifts (in our case, the shape parameter) along the boundaries of the contiguous regions. Again, estimation at ungauged locations between two homogeneous regions with significant difference in the regional shape
parameter will be a difficult decision. The method of ROI however circumvents some of the drawbacks of conventional RFA approach.

Finally, in our approach also we assumed spatial independence of the observations. This assumption will however not be true, especially since we have considered all the positive precipitation. We however expect the benefit of the regional approach to outweigh the consequences of ignoring the spatial dependence (Hosking and Wallis, 1988), especially since our interest is
on the marginal distribution only (Zheng et al., 2015). One interesting aspect also is to improve the method based on omega (see 3.2) to take into account both the margins and the dependence between sites.



*Author contributions.* AH performed the numerical computations and prepared the paper. All authors participated in the discussions, results analysis and paper writing.

*Competing interests.* The authors declare that they do not have any competing interests

*Acknowledgements.* This work has been funded by the Swiss Federal Office for Environment (FOEN) and the Swiss Federal Office for Energy through the project EXAR "Evaluation of Extreme Flood Events in the Aare-Rhine hydrological system in Switzerland"



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
