# Peer review of "Performance-based comparison of regionalization methods to improve the at-site estimates of daily precipitation"

_Hydrology and Earth System Sciences, 2021_

## Author Comment (AC2)

**Response to comments of Reviewer 2**

March 9, 2022

Thank you very much for your positive comments about our work. You can find below our response to your comments.

*General comments: The manuscript gives a nice overview of state-of-the-art methods for regionalization of precipitation, with a clear aim of comparing and evaluating these methods. Comparison strategy and evaluation criteria seem adequate and the different methods' strengths and drawbacks are well presented. The manuscript would however benefit from a clearer structure (especially of Chapter 3) and a more concise language (see below).*

**Response:** We are glad to hear that the reviewer find our manuscript interesting. We clarify below some of the points raised by the reviewer and will make the corrections in the revised manuscript.

**1 Specific Comments**

- *Chapter 1 Introduction: It could be stated more clearly what potential practical uses and benefits of such methods are, besides "modeling the whole range of positive precipitation*

  **Response:** We will clarify more on this in the revised manuscript. We will point out that modelling the whole range of precipitation has various practical applications. For instance in flood risk assessments, where stochastic precipitation generators are used to simulate long series of positive precipitations, extremes included, e.g. in [3] . The simulated precipitation is then used as input to conceptual hydrological models for simulation of long series of river discharge. Other practical applications are in the evaluation of numerical weather simulations or investigation of climatology of rainfall events as outlined by [1]

- *Chapter 2: Could you please comment on the histogram in Figure 1; why the large number of stations in Switzerland in 1900-09 for instance?*

  **Response:** The histogram in Figure 1 of the manuscript summarizes the data available for this study. It simply shows the number of gauges installed (or at least with records starting) in each decade. The large number of stations in Switzerland in 1900-09 simply reflects the fact that a lot of stations (more than 150) were installed within that decade. The same is highlighted on the map of the same figure, orange to dark red colored stations ($> 100$ years) are mostly located in Switzerland.

- *Chapter 3: It is unclear what the difference is between "methods of regionalization" and "regional models". I would prefer moving Table 1, and maybe the whole chapter 3.5 to the beginning, as an introduction to the models, and then describe details in the sub-chapters. Also consider adding a column to Table 1, naming the section where the model is described. You could also mark clearly in the text whenever one of the 5 models is described/introduced, since only number 5 has its own sub-chapter. I believe repetition, for instance in the summary at the end of chapter 3.2 and chapter 3.3, is a bit confusing. Could you rather base your description on the steps of the summary?*

  **Response:** We agree with the reviewer that the difference between the two is not necessarily very clear.

  To clarify, "methods of regionalization" as used in our work refers to the three different approaches to regionalization i.e. i) Regional frequency analysis based on the upper tail behaviour, ii) Region of influence approach (ROI), and iii) Spatial method using generalized additive model (GAM) forms.

  The "regional models" are the four models developed based on the the three outlined methods of regionalization as applied to the Extended Generalized Pareto Distribution (EGPD) of [4].

One model each based on the first and the last method of regionalization, and two models based on the ROI method. The fifth model is the EGPD without regionalization.

We will also review and revise the chapter to make the presentation of the approaches more clear and concise.

- *Chapter 5: The first five paragraphs (lines 337 - 356) describe the method and should be moved to a previous chapter. Would you please comment on the seasonal differences in Figure 3 (chapter 5.2).*

  **Response:** We agree that this paragraph describe the method and we will move it accordingly in the revised manuscript.

  The seasonal differences in Figure 3 of the manuscript is evident from the marked seasonality as well as the spatial variability of daily precipitation in Switzerland. We recall that the identification of the regions for each station is based on the homogeneity of the scaled extremes. The occurrence of these extremes and their quantity, which affects the test of homogeneity ([2]) depend on the season, and hence the observed seasonal differences.

**2 Technical corrections**

- *There are several examples of unclear language and wordy sentences. I have listed the most obvious ones below, but I recommend a full language review.*

  **Response:** We thank you for noticing these and we will do a full language review of the manuscript to identify and correct/rephrase them as well as others.

**References**

[1] Juliette Blanchet et al. "Mapping rainfall hazard based on rain gauge data: an objective cross-validation framework for model selection". en. In: *Hydrology and Earth System Sciences* 23.2 (Feb. 2019), pp. 829–849. ISSN: 1607-7938. DOI: 10.5194/hess-23-829-2019. URL: https://hess.copernicus.org/articles/23/829/2019/ (visited on 04/19/2021).

[2] G. Evin et al. "A regional model for extreme rainfall based on weather patterns subsampling". en. In: *Journal of Hydrology* 541 (Oct. 2016), pp. 1185–1198. ISSN: 00221694. DOI: 10.1016/j.jhydrol.2016.08.024. URL: https://linkinghub.elsevier.com/retrieve/pii/S0022169416305145 (visited on 08/25/2021).

[3] Guillaume Evin, Anne-Catherine Favre, and Benoit Hingray. "Stochastic generation of multi-site daily precipitation focusing on extreme events". en. In: *Hydrol. Earth Syst. Sci.* (2018), p. 18.

[4] Philippe Naveau et al. "Modeling jointly low, moderate, and heavy rainfall intensities without a threshold selection". en. In: *Water Resources Research* 52.4 (Apr. 2016), pp. 2753–2769. ISSN: 00431397. DOI: 10.1002/2015WR018552. URL: http://doi.wiley.com/10.1002/2015WR018552 (visited on 08/25/2021).

---

## Author Response (AR1)

**Response to comments of Reviewer 1 and 2**

April 12, 2022

**1 Reviewer 1**

Thank you very much for your positive comments about our work. You can find below our response to your comments in italic. We have also made the changes in the revised manuscript.

*This paper presents an interesting comparison of regionalization methods for rainfall frequency analysis. It is based on a large sample of 1176 daily stations in Switzerland and its neighbouring countries. Five models have been compared, based on local data (local EGPD) or regional data (Omega EGPD with an average shape parameter; ROI EGPD Full or Semi with ROI approach; GAM EGPD with parameters depending on covariates). The paper gives confirmation on the interest of regional approach vs local approach, and concludes that the GAM model is better for the upper tail, and the ROI model for the bulk of the distribution. The paper is clear and well written.*

**Response:** We thank you very for much again for the constructive feedback.

**1.1 General Comments**

- *I have one requirement on additional simulation and three minor recommendations. The authors decided to use a EGPD distribution, with the advantage of representing both the bulk of the distribution and the upper tail. As the conclusion is that no model is the best on the whole part of the distribution, I would be interested to see whether a GP-ROI distribution-regional approach performs on the upper tail, compared with a GAM EGPD model.*

  **Response:**

  As suggested by the reviewer, we ran the additional simulation where we regionalize, based on the method of Region of Influence (ROI), a Generalized Pareto Distribution (GPD). Following the same methodology outlined in our paper, we assessed the performance of this model on the upper tail using the FF and SPAN criteria. We show the result in Figure 1. For clarity, we include the boxplots of the other regionalization methods, as contained in Figure 5 of the main paper. Additionally we include two models:

  - ROI_GPD model: Here the GPD fitted on the POT of each station is regionalized according to the ROI method.
  - GAM_GPD model: Here a GAM model is used to regionalize the parameters of a GPD

  We comment that the performance of the GPD_ROI model in terms of both robustness and the reliably in the upper tail, is less compared to those of the GAM_EGPD model. The difference in performance is more pronounced in the case of robustness as measured by SPAN100, and in fact the GPD_ROI model has lower robustness compared to the other ROI models that are based on EGPD. Indeed, local GPD models (without any regionalization) have been shown to lack robustness [2]. For this reason, the fact that some stations have no neighbors identified (around 33% in the case of winter for example), it is not surprising that the low robustness of the GPD fitted on these stations will affect the overall regional SPAN100 score. The GAM_GPD model on the other hand shows high robustness and reliability in the upper tail.

  We remind the reviewer that although interesting, we choose not to show these results (GPD cases) in the paper because the paper aims at modeling the whole range of precipitation and not only the extremes.

**1.2 Minor recommendations**

- *Line 152. The first time, explain PAM acronym (Partitioning around medoids)*

  **Response:**

[Figure]

Figure 1: Criteria applied on the upper tail for each season. Left: Robustness of the local EGPD and the six candidate models, as measured by the SPAN criteria. The stability is measured with respect to a 100-year return level estimate. Each boxplot contain 50 values. Right: Reliability in prediction of the maxima as measured by the FF criteria, each boxplot contain 100 values.

Thank you very much for this correction. We have included the meaning of the acronym, see line **158**

- *Figure 3 - Histogram. It could be interesting to add (for each class of radius) the mean number of stations belonging to the ROI*

    **Response:** We have modified the Figure to include the average number of stations identified for each class of ROI. See the caption of **Figure 3**

- *Line 420. "The plot of the right of Figure 5"*

    **Response:**

    Thank you again for this correction. We have effected the correction. See line **412**

**2 Response to comments of Reviewer 2**

Thank you very much for your positive comments about our work. You can find below our response to your comments.

*General comments: The manuscript gives a nice overview of state-of-the-art methods for regionalization of precipitation, with a clear aim of comparing and evaluating these methods. Comparison strategy and evaluation criteria seem adequate and the different methods' strengths and drawbacks are well presented. The manuscript would however benefit from a clearer structure (especially of Chapter 3) and a more concise language (see below).*

**Response:** We are glad to hear that the reviewer find our manuscript interesting. We clarify below the points raised, and we have made the corrections in the revised manuscript.

**2.1 Specific Comments**

- *Chapter 1 Introduction: It could be stated more clearly what potential practical uses and benefits of such methods are, besides "modeling the whole range of positive precipitation*

  **Response:** We have clarified more on this in the revised manuscript. Please refer to line **35** to line **39** where we included the following text:

  *Modeling the whole range of precipitation has various practical applications. For instance in flood risk assessments, where stochastic precipitation generators are used to simulate long series of positive precipitations, extremes included [3]. The simulated precipitation is then used as input to conceptual hydrological models for the simulation of long series of river flows. Other practical applications are in the evaluation of numerical weather simulations or investigation of the climatology of rainfall events as outlined by [1]*

- *Chapter 2: Could you please comment on the histogram in Figure 1; why the large number of stations in Switzerland in 1900-09 for instance?*

  **Response:** The histogram in Figure 1 of the manuscript summarizes the data available for this study. It simply shows the number of gauges installed (or at least with records starting) in each decade. The large number of stations in Switzerland in 1900-09 simply reflects the fact that a lot of stations (more than 150) were installed within that decade. The same is highlighted on the map of the same figure, orange to dark red colored stations ($> 100$ years) are mostly located in Switzerland.

- *Chapter 3: It is unclear what the difference is between "methods of regionalization" and "regional models". I would prefer moving Table 1, and maybe the whole chapter 3.5 to the beginning, as an introduction to the models, and then describe details in the sub-chapters. Also consider adding a column to Table 1, naming the section where the model is described. You could also mark clearly in the text whenever one of the 5 models is described/introduced, since only number 5 has its own sub-chapter. I believe repetition, for instance in the summary at the end of chapter 3.2 and chapter 3.3, is a bit confusing. Could you rather base your description on the steps of the summary?*

  **Response:** We agree with the reviewer that the difference between the two is not necessarily very clear.

  To clarify, "methods of regionalization" as used in our work refers to the three different approaches to regionalization i.e. i) Regional frequency analysis based on the upper tail behaviour, ii) Region of influence approach (ROI), and iii) Spatial method using generalized additive model (GAM) forms.

  The "regional models" are the four models developed based on the the three outlined methods of regionalization as applied to the Extended Generalized Pareto Distribution (EGPD) of [4]. One model each based on the first and the last method of regionalization, and two models based on the ROI method. The fifth model is the EGPD without regionalization.

  We have included this clarification in the beginning of chapter 3, see **line 118** to **line 122**. Additionally, we have reorganized the chapter for more clarity. Chapter 3.1 presents the marginal distribution, chapter 3.2 presents the methods of regionalization, and chapter 3.3 describes the regional models built. We also added a column in Table 1, as suggested by the reviewer, to name the section where each particular model is described.

- *Chapter 5: The first five paragraphs (lines 337 - 356) describe the method and should be moved to a previous chapter. Would you please comment on the seasonal differences in Figure 3 (chapter 5.2).*

  **Response:** We agree that this paragraph describe the method and we have moved it to section 3.3.1. See line **190** to **193**

  The seasonal differences in Figure 3 of the manuscript are due to the marked seasonality and the spatial variability of daily precipitation in Switzerland. Recall that the identification of the regions is based on the homogeneity of the scaled extremes. The occurrence of these extremes and their quantity, which affects the test of homogeneity ([2]), depends on the season. Hence the observed seasonal differences. We have included this in the revised manuscript, see line **360** to **363**.

**2.2 Technical corrections**

- *There are several examples of unclear language and wordy sentences. I have listed the most obvious ones below, but I recommend a full language review.*

  **Response:** Thank you for noticing these, and we did a full language review as suggested. Below are the lines where we corrected the specific cases pointed out by the reviewer.

L1: performances – performance: **corrected**, see line **1**.

L15: ”these events” points back to the previous sentence, where you also write ”these events”. Please specify which events you are referring to: **corrected**, see line **15**.

L19: Missing a comma after ”signal”: **corrected**, see line **19**.

L28: What does ”This” refer to?: ”This” refers to ”Mixture models”. we have rephrased the sentence. see line **28**.

L43: Remove quotation marks around ”regional”: **removed**, see line **50**.

L58: Remove ”though”: **removed**, see line **65**.

L68: Add commas after ”authors” and after ”study”: **added**, see line **75**.

L72: ”... also using GEV compared the ...” – ”... also used GEV to compare the...”: **rephrased**, see line **79**.

L78: ”...we consider a model, the EGDP ...” – ”...we consider the EGDP model ...”: **rephrased**, see line **85**.

L81: Remove ”by”: **removed**, see line **88**.

L93: Remove ”by”. ”recorded at” – ”from”: **corrected**, see line **100**.

L96: Add ”While” before ”the main study area...”: **added**, see line **103**.

L98: ”the whole” – ”all”: **replaced**, see line **105**.

L102: Please rephrase “It is also marked by...”: **rephrased**, see line **108**.

L104: Please rephrase “The Ticino stands generally as the...”: **rephrased**, see line **111**.

L107: Please rephrase. “necessitates...” does not follow from “Resulting from...”: **rephrased**, see line **114**.

L115. Should it be “precipitation”, not “rainfall”?: **replaced**, see line **124**.

L117: Please define “positive precipitation”. Non-zero?: **defined**, see line **125**.

L118: Remove “though”: **removed**, see line **126**.

L256: Remove “method”: **removed**, see line **256**.

L405: Remove “for this model”:**removed**, see line **397**.

L420: Add “Figure” before “5”: **added**, see line **412**.

L427: Please rephrase. For instance “In Figure 6 we also show the 100-year return level...”: **rephrased**, see line **418** to **422**.

L429: Remove “in comparison to the other regions”: **removed**, see line **418** to **421**.

L430: Remove “in comparison to the other seasons”: **removed**, see line **421**.

L441: Suggest to rephrase to “We based our comparison on criteria that measure ...”: **rephrased**, see line **430**.

L449: “estimate” ”-” “estimation”: **replaced**, see line **438**.

L471: Remove the first “Of”: **removed**, see line **460**.

**References**

[1] Juliette Blanchet et al. “Mapping rainfall hazard based on rain gauge data: an objective cross-validation framework for model selection”. en. In: *Hydrology and Earth System Sciences* 23.2 (Feb. 2019), pp. 829–849. ISSN: 1607-7938. DOI: 10.5194/hess-23-829-2019. URL: https://hess.copernicus.org/articles/23/829/2019/ (visited on 04/19/2021).

[2] G. Evin et al. “A regional model for extreme rainfall based on weather patterns subsampling”. en. In: *Journal of Hydrology* 541 (Oct. 2016), pp. 1185–1198. ISSN: 00221694. DOI: 10.1016/j.jhydrol.2016.08.024. URL: https://linkinghub.elsevier.com/retrieve/pii/S0022169416305145 (visited on 08/25/2021).

[3] Guillaume Evin, Anne-Catherine Favre, and Benoit Hingray. “Stochastic generation of multi-site daily precipitation focusing on extreme events”. en. In: *Hydrol. Earth Syst. Sci.* (2018), p. 18.

[4] Philippe Naveau et al. “Modeling jointly low, moderate, and heavy rainfall intensities without a threshold selection”. en. In: *Water Resources Research* 52.4 (Apr. 2016), pp. 2753–2769. ISSN: 00431397. DOI: 10.1002/2015WR018552. URL: http://doi.wiley.com/10.1002/2015WR018552 (visited on 08/25/2021).